# Considerations in implementation of social risk factor screening and referral in maternal and infant care in Washington, DC: A qualitative study

Jason Brown[1], Naheed Ahmed[1], Matthew Biel[2,3], Loral Patchen[4,5], Janine Rethy[6], Angela Thomas[1], Hannah Arem[1,7] *

1 Medstar Health Research Institute, Healthcare Delivery Research, Washington, DC, United States of America, 2 Department of Psychiatry, Georgetown University School of Medicine, Washington, DC, United States of America, 3 Department of Psychiatry, MedStar Georgetown University Hospital, Washington, DC, United States of America, 4 MedStar Washington Hospital Center, Women and Infant Services, Washington, DC, United States of America, 5 MedStar Washington Hospital Center, Obstetrics/Gynecology, Washington, DC, United States of America, 6 Department of Pediatrics, MedStar Georgetown University Hospital, Washington, DC, United States of America, 7 Department of Oncology, Georgetown University School of Medicine, Washington, DC, United States of America

* Hannah.Arem@medstar.net

**Data Availability Statement:** Because of the small sample size among employee participants and the limited number of clinics participating, it would be

## Abstract

### Background

The District of Columbia (DC) has striking disparities in maternal and infant outcomes comparing Black to White women and babies. Social determinants of health (SDoH) are widely recognized as a significant contributor to these disparities in health outcomes. Screening for social risk factors and referral for appropriate services is a critical step in addressing social needs and reducing outcome disparities.

### Methods

We conducted interviews among employees (n = 18) and patients (n = 9) across three diverse, urban clinics within a healthcare system and one community-based organization involved in a five-year initiative to reduce maternal and infant disparities in DC. Interviews were guided by the Consolidated Framework for Implementation Research to understand current processes and organizational factors that contributed to or impeded delivery of social risk factor screening and referral for indicated needs.

### Results

We found that current processes for social risk factor screening and referral differed between and within clinics depending on the patient population. Key facilitators of successful screening included a supportive organizational culture and adaptability of more patient-centered screening processes. Key barriers to delivery included high patient volume and limited electronic health record capabilities to record results and track the status of internal and community referrals. Areas identified for improvement included additional social risk factor

easy to identify participants based on how they describe their roles in transcripts. Thus we propose only releasing transcript components to those with reasonable request rather than posting them publicly or allowing unrestricted access. Requests to access the data should be sent to research@medstar.net or 301-560-7300 and should reference the scientific interest in the data as well as what is being requested.

**Funding:** Funding was provided by the A. James and Alice B. Clark Foundation, titled Safe Babies Safe Moms (SBSM) project. The funder was not involved in the design of the study and collection, analysis, and interpretation of data, nor in writing the manuscript.

**Competing interests:** The authors have declared that no competing interests exist.

assessment training for new providers, patient-centered approaches to screening, improved tracking processes, and facilitation of connections to social services within clinical settings.

## Conclusion

Despite proliferation of social risk factor screeners and recognition of their importance within health care settings, few studies detail implementation processes for social risk factor screening and referrals. Future studies should test implementation strategies for screening and referral services to address identified barriers to implementation.

## Background

Black women have 1.4–2.5x higher rates of maternal mortality, pre-term birth, and low birth weight infants than similar non-Hispanic White women [1–6]. The District of Columbia has some of the highest rates of maternal mortality in the U.S. From 2014–2018 the 5-year pregnancy-related mortality rate was 44.0 deaths per 100,000 live births compared to 28.4 deaths per 100,000 live births across the US [7]. This burden is unequally distributed, with 90% of pregnancy-related births experienced by Black birthing individuals [7].

Social determinants of health (SDoH) are defined by the World Health Organization as "the conditions in which people are born, grow, live, work, and age," and which are "shaped by the distribution of money, power and resources" (e.g. neighborhood, income, education, social connections, and access to healthcare) [1, 5, 8–10]. SDoH encompass "upstream" factors or systemic issues related to public policies, which in turn affect downstream social factors such as physical living or working environments [11]. These SDoH also directly impact an individual's social risk factors associated with poorer health outcomes including housing instability, lack of transportation for medical appointments, or food insecurity [12–14]. It is estimated that clinical care, including access to care and quality of care, accounts for an estimated 20% of health outcomes, while other factors, including SDoH and individual social risk factors, explain 80% of health outcomes [15]. Addressing unmet social needs, or those social risk factors for which patients want to seek help, thus could contribute to improving health outcomes and reducing disparities, especially in areas such as DC where maternal mortality rates and disparities remain high.

Despite growing recognition from professional societies, researchers, clinicians, and payers of the importance of screening for social risk factors and addressing identified social needs [16–18], there is a gap between what is recommended and what is implemented. There is also limited published research on implementation of social risk factor screeners and delivery of appropriate patient referrals in complex healthcare settings, where processes related to social risk factor screening and referral vary widely. A recent systematic review on social needs screening highlighted the bias in existing studies in assessing impact on health outcomes, calling for higher quality studies to assess impact on outcomes of interest [19]. The COVID-19 pandemic not only exacerbated social needs like food insecurity and housing instability, but has also affected the capacity of clinical staff to screen and refer for these needs. Due to the pandemic community-based organizations have also encountered challenges in providing for increased volume of needs or pivoting towards virtual services in order to protect staff and clients.

Furthermore, to our knowledge there are no existing publications using the Consolidated Framework for Implementation Research (CFIR) to describe organizational factors

contributing to social risk factor screening and referral implementation [20–23]. We selected the CFIR framework to demonstrate potential approaches for other clinical care settings considering implementing or modifying social risk factor screening and social needs referral processes [20–23]. Given the complexity of social risk factor screening and referral for unmet social needs in sites with different assessment tools, patient volumes, patient populations, and workflows, we set out to describe existing processes across three specialty clinics and one community based organization (CBO) participating in a project to reduce maternal and infant health outcome disparities in Washington, DC. We thus describe social risk factor screening and social needs referral barriers and facilitators in real-world settings to understand existing processes and identify potential areas for improvement. Our study protocol was reviewed and approved by the MedStar Health Research Institute Institutional Review Board (STUDY00003156). There were no conflicts of interests from the authors or study team.

## Methods

### Sample

This study was conducted in December 2020-March 2021 in the first year of a 5-year initiative to reduce disparities in maternal and infant health outcomes in Washington, DC [24]. The initiative, now starting its third year, includes more than a dozen interventions across multiple clinical specialties within an urban healthcare system and partner CBOs to support women of reproductive age and children up to the age of 3, with a particular emphasis on reducing disparities in morbidity and mortality outcomes.

Our sample was recruited using convenience sampling. Project leads from each of the four involved clinics were asked to identify at least one employee from each of the following two levels: front line staff (front desk staff, clinic managers, social worker), clinical advanced practitioners (nurse practitioner, physician assistant, medical doctor, midwife, etc.). Employee interview participants were healthcare system or community partner organization employees (18 years of age or older) who worked at one of the following clinics: OB-GYN, Family Medicine, or Pediatrics. We also included employees of a CBO involved in the 5-year project that provides maternal and infant health services. Although more community organizations are involved in the project over the five years, we were only able to conduct interviews with the partner who had an executed agreement in place in the first year. Response rates for employee participants contacted for this study were 66%, 83%, 100%, and 100% for each of the three clinics and the CBO, with an overall response rate of 86% (n = 18/21).

Each of the clinics was also asked to invite at least three patients who were eligible for a SDoH screening within the prior three months for interviews to better understand the patient experience. Eligible patients were adults (18 years of age or older), English speaking, pregnant or postpartum women, or those considering pregnancy who had at least one care encounter over the prior three months. No patients were recruited from the CBO due to the organization's plans to also interview patients about other maternal health related questions and a desire to combine all interviews into a single process to lessen the burden on patients; the CBO expressed an interest in incorporating similar questions into future interviews. Response rates for patient interviews were lower compared to employees and were 23%, 60%, and 100% for each of the three clinics with an overall response rate of 42% (n = 9/21). Patients who did not participate did so for one of the following reasons: Spanish language was primary language (n = 2), did not answer calls (n = 6) or email (n = 1), number was not in service (n = 2), or failed to join the phone interview (n = 1) at the scheduled time. Of the n = 9 included patients n = 7 identified as African American, and n = 2 identified as "Other". Eight of the nine reported public insurance. All participants were female and ranged in age from 21 to 37 years old.

Consent forms were emailed to all individuals prior to each interview. Following phone-based review of study details, verbal consent to participate in study was obtained at the start of each interview. Patients were compensated $50 for their time and participation in the form of a mailed check. Employee participants did not receive any compensation for their participation.

## Data collection

The CFIR can be used to systematically capture multi-level influences and contextual factors that facilitate or impede implementation processes [25]. The CFIR is widely used in implementation science to capture organizational context; thus our interview guide largely captured organizational characteristics impacting intervention implementation [26–28]. Questions were based on five CFIR domains of intervention characteristics, characteristics of individuals involved, inner setting (organizational culture and relative priority), and outer setting (patient needs and resource availability/quality), and process [27, 28]. Within each domain only those constructs deemed relevant to our study were included (Table 1). We further developed our interview guide to understand current social risk factor screening/referral processes, barriers and facilitators of completing social risk factor screening/referrals. The patient interviews focused more on process and experiences with social risk factor screening and referrals than the organizational CFIR constructs. The employee constructs are described in Table 1, along with the corresponding interview question(s).

All interviews were conducted using WebEx videoconferencing. Interviews included one interviewer, one interviewee, and when available, one notetaker. Interviews were video recorded where the participant had video available, or audio recorded if no video was available. Interviews with employees were 60 to 90 minutes in duration and patient interviews were 30 to 45 minutes in duration. Transcription was done using the NVivo (version 1.5.1) software automated transcription module. Interview transcripts were quality checked manually by at least one study team member prior to coding.

## Data analysis

We used a deductive thematic content analysis based on *a priori* CFIR constructs, with additional codes added as needed. Initial codes were generated by two coders (HA and NA) with qualitative/mixed-methods research backgrounds. Each of the three coders (HA, NA, JB) reviewed two patient and employee transcripts to resolve coding differences and reach a consensus on the coding framework, and application of codes to initial themes based on CFIR constructs prior to splitting up remaining transcripts. New themes that emerged were discussed with the group so that codes could be applied similarly between the three reviewers. Interviews were independently coded by two coders (patient interviews: JB and NA, employee interviews HA and JB).

## Results

We conducted 27 semi-structured in-depth interviews (n = 18 employee and n = 9 patient) between December 2020-March 2021 (Fig 1). Patients ranged in age from 21–37 years old and all were female. Seven reported African American race and two reported "other". Eight out of nine included patients subscribed to public insurance. We collected the screeners used in each of the clinics at the time of interviews and detail the social needs domains covered in the respective screeners in S1 Table. It is important to note that providers mentioned other domains coming up in discussion, but we included only those which were specified in screeners in the Table.

**Table 1. Consolidated framework for implementation research constructs used in employee interviews [35].**

| Construct | Short Description | Related interview question |
|---|---|---|
| Domain 1: Intervention Characteristics | | |
| Relative advantage | Stakeholders' perception of the advantage of implementing the intervention versus an alternative solution. | If resources were not an issue, what would you change in the SDoH screening and referral process at present? |
| Adaptability | The degree to which an intervention can be adapted, tailored, refined, or reinvented to meet local needs. | Do you have suggested changes for how patients are screened for SDoH to work more effectively and to increase uptake? |
| Design Quality and Packaging | Perceived excellence in how the intervention is bundled, presented, and assembled. | What supports, such as online resources, shared folders, or a toolkit, are available to help you screen and refer for SDoH? How do you access these materials? |
| | | What is your perception of the quality of these materials? |
| Cost | Costs of the intervention and costs associated with implementing the intervention including investment, supply, and opportunity costs. | What are the costs to SDoH screening in terms of staff, time, or other resources? |
| | | What about costs for referrals or following up on referrals? |
| Domain 2: Outer Setting | | |
| Patient Needs and Resources | The extent to which patient needs, as well as barriers and facilitators to meet those needs, are accurately known and prioritized by the organization. | How well do you think that the current SDoH screening and referral process meets the needs of the patients and families served by your organization? |
| | | In what ways does intervention meet their needs? E.g. improved access to services? Reduced wait times? Help with self-management? Reduced travel time and expense? |
| | | What barriers will the individuals served by your organization face to screening and referring for SDOH? |
| | | Have you heard stories about the experiences of participants with SDOH referrals? Can you describe a specific story? |
| Domain 3: Inner Setting | | |
| Structural Characteristics | The social architecture, age, maturity, and size of an organization. | What infrastructure changes would you suggest to better screen and refer for SDOH? |
| | | Changes in the time you have with patients? Changes in formal policies? Changes in information systems or electronic records systems? Other? |
| Culture | Norms, values, and basic assumptions of a given organization. | How do you think your organization's culture (general beliefs, values, assumptions that people embrace) affect your ability to screen and refer for SDOH? |
| | | Can you describe an example that highlights this? |
| Implementation climate: | The absorptive capacity for change, shared receptivity of involved individuals to an intervention and the extent to which use of that intervention will be rewarded, supported, and expected within their organization. | |
| • Relative Priority | Individuals' shared perception of the importance of the implementation within the organization. | How do you juggle competing priorities in your own work? How do your colleagues juggle these priorities? |
| • Organizational Incentives and Rewards | Extrinsic incentives such as goal-sharing awards, performance reviews, promotions, and raises in salary, and less tangible incentives such as increased stature or respect. | What kinds of incentives are there to help ensure that screening and referral for SDOH is successful? What is your motivation for wanting to help ensure the implementation is successful? |
| Domain 4: Characteristics of Individuals | | |
| Self-efficacy | Individual belief in their own capabilities to execute courses of action to achieve implementation goals. | How confident are you that you are able to successfully complete screening and referral for SDOH at this time? |
| | | What gives you that level of confidence (or lack of confidence)? |
| | | How confident do you think your colleagues feel about screening and referring for SDOH at this time and what gives them that level of confidence (or lack of confidence)? |
| Domain 5: Process | | |

(*Continued*)

**Table 1.** (Continued)

| Construct | Short Description | Related interview question |
|---|---|---|
| Executing | Carrying out or accomplishing the implementation according to plan. | Please tell me about how you currently screen for SDoH in your clinic. Who is involved in this process (who administers it), how often do patients receive screening, and who reviews screening? |
| | | What facilitates successful completion of SDoH screening/referrals? |
| | | What are the biggest barriers to screening/referrals? |

### Social risk factor screening processes- employee perspectives

Both the baseline and the ongoing adapted processes for social risk factor screening varied significantly across clinic sites (Table 2). One clinic reported using hard copies of screening questionnaires self-administered by patients in the waiting room or during their visit, which was then reviewed by the behavioral health team. Another clinic had patients fill out screening questionnaires electronically prior to visits either at home or in the waiting room. However, this required significant administrative support, asking front desk staff to reach out to patients missing screening information by phone to address missing or incomplete forms. A third clinic had the provider administer social risk factor questions during the clinic visit and entered responses directly into the patient's electronic medical record. Only one clinic used a social risk factor screener embedded in the electronic medical record, although the other clinics had plans to integrate forms as part of the 5-year disparities initiative. Only pediatric visits included social risk factor screening questions related to domains like transportation, food insecurity and housing universally as part of standard of care. In other clinics screening questions for adult populations varied by specialty and even within clinics depending on whether a woman was pregnant or not. Most of the clinics seeing prenatal women included universal screenings around domains such as depression, substance abuse, and intimate partner violence prior to the 5-year initiative but not around domains such as transportation or food insecurity; still, some of these issues were discussed by providers in routine care and were referred to

**Fig 1. Interview participants by department and role.**

**Table 2. Comparison of screening and referral across included sites.**

| Subthemes | Site 1 | Site 2 | Site 3 | Site 4 |
|---|---|---|---|---|
| **Screened population at the time of interviews** | Children at all well visits | Children at all well visits | New obstetrics (OB) population | New OB population |
| **Planned goals for screening expansion** | n/a | Screening new OB and third trimester | Screening third trimester and post-partum | n/a |
| **Screening tool** | Internally developed in conjunction with community partner | Bright Futures for pediatrics | Internally developed for pregnant and post-partum patients | OB authorization form for DC Medicaid and PRAPARE tool |
| **Screening delivery modality at the time of interviews** | TONIC-based questionnaire | Documented directly in EHR-embedded form | Paper forms delivered by behavioral health navigator adjacent to the new OB visit | Paper forms completed on site with patient adjacent to new OB visit, which is later entered into a smart form in the EHR |
| **Who delivers screening** | Self-administered questionnaire with follow up by office staff by phone if patient doesn't respond | Physician provider during clinical visit | Behavioral health team | Prenatal care coordinator |
| **Planned changes for screening delivery** | Better integrating TONIC into EHR | Powerform being built for screening new OB | Powerform was being built at the time of interviews; form is being shortened | Better integrating smart form into practice |
| **Who links patients with identified needs to resources** | Shared folder with flyers that provider can distribute; follow up by social worker | Behavioral health navigator/ social worker | Behavioral health navigator/ social worker | Prenatal care coordinator |

social workers and documented in chart notes even if they were not universally screened for using a standardized form in the medical record.

## Social risk factor referral processes- employee perspectives

Across the clinic sites referral processes for identified social needs varied in response to multiple factors including patient age, patient volume, and availability of social workers or navigators to address identified needs. Furthermore, between clinics providers addressing social needs outlined different processes for conducting referrals, different resource information management methods, and different levels of assistance they provided to connect patients with resources. Resource delivery ranged from providing handouts or contact information for available services to case management with ongoing follow-up with a social worker and/or behavioral counseling for mental health. These responses were based on both severity and complexity of the identified needs, as well as availability of resources and patient volume. Furthermore, some clinics opted for trying to teach patients how to navigate the system on their own, while others sought to facilitate connections to community programs on behalf of the patient. Tracking processes for referrals varied but were largely managed by the social workers using systems outside the medical record (e.g. excel spreadsheet), with providers informed through group huddles or notes in the medical record.

## Social risk factor screening and referral processes- patient perspectives

In interviews with patients, some participants described ways that the social needs support was delivered in conjunction with their medical care, while others did not recall having such conversations with their medical providers. One notable finding was that patients were receptive of having these conversations in a doctor's office, but did not differentiate about who was asking which questions. However, wait time surfaced as a barrier.

> *It went very good, because to be honest, I really didn't know the doctors asked about those type of things.–Patient 7*

*I'm not sure who I spoke with. One of the nurses there was just asking questions like am I able to provide food for my family and everything. And my answer was yeah, I could make ends meet...* -Patient 10

*I only met with her, like once, I was supposed to meet with her for a second time, but I had to go. So I left because she had me waiting too long*–Patient 2

Key resources that were accessible to patients included food benefits, but there was a noted gap around identifying or supporting housing issues.

*I used the food banks and definitely the diapers and whatever clothing things they had in the area, you know, try to go past as much as I can. Oh also and the farmers market, they gave me really good farmers market information because now covid and now that farmers market now taking WIC. They also gave me WIC services too they gave me my WIC referrals... I did not know that I had access to all of these things in my area. So it wasn't necessarily them, doing anything for me, it was more so just the information in general, they informed me on a lot of things that was going on in my area that I did not know.* -Patient 3

*Well, they always, every time you go or whatever, they check on your mental health and make sure and I ain't never hear them say anything about if you need help with paying rent or anything. But definitely resources for food or whatever or any social services things.... It went very good, because to be honest, I really didn't know the doctors asked about those type of things... But yeah, they'll give you resources. They'll reach out to the social worker*–Patient 7

*so just in case because I'm only getting public assistance. By me not having shelter or me looking for shelter. I don't know if I'm going to have to be in a room or rent an apartment... well, rent hotels, until they find something, because I really don't want to be in the streets—*Patient 2

Patients who received information about a service alone did not necessarily connect to the resource, especially if it was not a desired solution, but also did not suggest any changes to the referral process.

*I mean I have the paper and I should be looking at it now that I think about it, I forgot I had that paper, but most of it is shelters and stuff. I'm trying to avoid going into a shelter because the pandemic, you know, a lot of people I don't like to be around a lot of people.*–Patient 2

*I really don't have any problems, I really don't have any issues, I have a great team. They're really good with keeping up on everything. I have multiple people reaching out to me. I'm pretty satisfied with my services that I'm receiving.*–Patient 3

## Barriers and facilitators of social risk factor screening and referrals

Barriers, facilitators, and potential improvements to screening and referral were explored across the five CFIR domains among providers and are described below and in Table 3. Suggestions for improving screening processes included conducting screening prior to visits, training clinic staff on screening administration (including patient-centered delivery), and streamlining or shortening questionnaires. Potential suggestions for improving referral processes included meeting with the social workers or directly connecting patients to a resource during the clinical visit rather than providing referral information after a visit, additional translated resources, and using electronic tracking forms for identifying needs and assessing

**Table 3. Summary of social risk factor screening barriers, facilitators, and potential improvement suggestions.**

| Screening step | Barriers | Facilitators | Potential Improvements |
|---|---|---|---|
| **Conducting screening** | Time needed to complete screening/Length of screening | Screeners filled out prior to at home or in waiting room | Streamlined or shortened questionnaires |
| | Patient volume relative to staffing levels | Screeners filled out prior to at home or in waiting room | Conducting more screening prior to patient arrival or in waiting room prior to patient visit |
| | Limited physical space to meet with patients | Screeners filled out prior to at home or in waiting room | Screeners filled out prior to at home or in waiting room |
| | Screening for some needs which can't be addressed or intervened on | Setting patient visit expectations | Additional training on screening administration |
| | Patient's expectations of visit length/content | Patient incentives (e.g. providing free diapers or another tangible benefit) | |
| | Language barriers | Interpreter/Translation services | Additional translated resources in Spanish, Amharic, and other languages. |
| | Patient discomfort with screening questions | Welcoming clinic culture to make patients feel comfortable | More patient-centered screening to increase patient comfort during screening |
| **Following up on positive screens** | Burden of documentation in multiple places (e.g. medical notes and individual tracking in excel) | Electronic tracking forms | Increased EHR flagging of positive SDoH needs from screening |
| | Lack of provider compensation when additional time is required to better ascertain need or facilitate connection with another provider or resource | Triage map on who to direct where (e.g. when provider can provide resource vs when social worker is needed) | Co-located community-based resources |

referral follow up. Still, meeting with social workers during a clinic visit was limited by patient volume and patient preferences around length of the appointment.

## Domain 1: Intervention characteristics

**Adaptability.** Adaptability was paramount to delivering social risk factor screening and referral. Particularly during the COVID-19 pandemic, specialty clinic processes evolved to meet the changing needs of their patients and instituting new safety protocols. For instance, in order to minimize patient crowding in the waiting room, one of the specialty clinics that used to have patients fill out the questionnaire on a tablet in the waiting room pivoted to sending the questionnaire by email or text ahead of time, and further split well visits into two components: one by telehealth to complete the patient history and discuss questionnaires including social risk factors, and the second, the following day, as a physical exam allowing the team time to discuss how to address any identified needs.

Employee participants also mentioned the importance of being able to adapt sensitive social risk factor screening questions to facilitate comfort with material or to help patients better understand certain questions. These findings highlight the importance of understanding the relationship between fidelity and adaptability, and how modifications to delivery affect screening and referral outcomes from both patient and provider perspectives. Feedback from providers also highlights the patient experience and difference between a conversation and a highly regimented screening.

*"Every so often there are people who are like, why are you asking this? And I feel comfortable and confident sharing like it's a standardized form. You can answer as much or as little as you want to, but we ask because we know more than your physical health impacts your pregnancy. But I feel bad about the questions. Like there's a question about if someone has met with a probation officer in the last year. I don't understand how having a probation officer or*

*not impacts the woman having a healthy. . . So I just change it when I say it. . ."*Front line
employee 15

*". . . I sometimes may reword them. So it's just a little easier for them to understand. So I
think there's a question about if they're having problems with WIC and food benefits and
those, and SNAP benefits. I just reword and just say, hey, are you having any issues with this
or that? Just something specifically. So it's not you know, an extremely formal question it's
more of a conversation with them."–Front line employee 6*

**Design quality & packaging.** Many employee participants described improvement over
time in the screening questionnaires, but some felt that the current questionnaires still might
be missing key areas of need or that the survey length was too long, particularly since it is col-
lected at the same time as numerous other questionnaires. While researchers distinguish these
domains in screening, clinicians may see the screenings as a package that needs to be covered
all together.

*". . . we now have a seven-page questionnaire with a lot of time and . . . input by . . . people
who are very well versed in this sort of thing. And just building up seven pages of very, very
detailed questions about safety, about housing, food, transportation, thoughts about becoming
a parent or being a parent. . .So it's much more expansive, much more comprehensive. . ."*–
Advanced practitioner employee 12

*"So I've kind of even paused because when you think about screening fatigue or the misunder-
standing of the screening and how much help the families need, I can't say with 100 percent
certainty that it's always the well-being screening specifically that the family's having a hard
time with. . .And then I think the thing that kind of like the snowball effect, though, is what if,
for example, if it's a mental health screening, once that parent is fed up with the mental health
screening, it's hard to get them to say, OK, well, let's stop that one and please read the well-
being. . .it's hard to say because it's being done with other screening."–Front line employee 1*

Additionally, referral resources are currently not shared across clinics within the healthcare
system or broadly to different providers within clinics and are instead managed by the social
worker in each clinic. This can lead to missed opportunities for resource sharing and potential
loss of knowledge when employees leave or take time off.

*". . . It was just resource knowledge I had in my head, but I didn't just keep it in my head. . .
[but] she [supervisor] asked me to save things on the Shared drive. So I started building up the
resources. So I pretty much created all the folders for the social work folder in the Shared
drive. . . It's for [this practice], but I don't know who has access."* -Front line employee 11

There was limited interest from social workers in having outside-curated or shared
resources between clinics, or even within clinics due to the perception of the unique needs of
the patients served, the care teams, and the difficulty maintaining shared databases and knowl-
edge sharing opportunities across sites. During the pandemic resources were also changing
constantly and social workers emphasized the importance of their knowledge of the changing
offerings and need to continually revisit what was available for patients.

**Costs.** Employees reported that screening necessitates dedicated clinical staff to adminis-
ter questionnaires (e.g. administrative staff, patient coordinators), clinic space to privately
administer questionnaires, and follow up to address needs included additional behavioral/

mental health services and increased number of social workers. Clinics have to balance between Relative Value Units (RVUs) for reimbursement and additional screenings or referrals that are not reimbursed. A main priority under the 5-year initiative was to hire and onboard additional staff across the clinics delivering services to the target population, including integrating mental health services within each clinical site.

> *"And the only way that, you know, the way the doctors make money is by seeing patients. So anything that occurs outside of a visit is not really supported. So that's kind of all extra work that is not ever compensated. So, I mean, that's the reason why I think we don't have a social worker at [other location] as much because there's no need. It's just that there's no funds to support that because you don't get paid for that. It's the only the doctors and patients that generate any income. And I mean, I think that's one of the main issues with health care. Like, there's so much that goes on outside of a patient visit that needs to be paid for. And yes, cost is a huge, huge issue."*–Advanced practitioner employee 2

Improvements in clinic location and space were also in process at the time of the interview, which may alleviate some of the space-related screening and referral limitations.

## Domain 2: Outer setting

**Patient needs & resources.** A high proportion of patients in the included clinics had complex needs. Participants often expressed the importance of trying to identify patients at high-risk for adverse clinical outcomes or with multiple or severe social needs. Extra time to screen and address the needs of these patients was described as important but challenging due to patient volume or limited time allowed to spend with patients. This tradeoff between addressing the needs of complex patients well and capturing all patients with screenings was described by several employees.

> *"Twenty five percent maybe have no real identifiable needs. Twenty five percent have some small need. And then the other 50 percent have maybe some sort of significant need that may or may not need case management specifically, but, you know, have something. And then about 15 percent. Yeah. With acute mental health needs. Safety issues, homelessness, stuff like that."*–Employee 12

Interview participants were asked how well social risk factor screening and referral processes were able to address patient needs. Most patient and employee participants felt that the resources provided were helpful but additional resources, stronger community partnerships, or additional support in connecting patients to resources and services more directly would significantly improve impact for patients.

> *"Because looking at this, we have other, we have a lot of different family support, infant supplies, areas in the district and housing and stuff like that. But I do think we need more liaisons from each of them so that we are able to provide for our patients properly."* -Front line employee 14

Providers also mentioned a need to make sure that patients understood the role of social workers and were given a choice in whether they want to engage with their services, rather than assuming all people will want help even if they have an identified social need. Specifically, providers described how some patients might be averse to discussing social needs within the medical setting.

*"...a lot of our patients have prior experiences with maybe social workers, case workers, they're low-income, they know about different services and governmental programs. So, when they do hear that someone's asking questions, they get comfortable. But sometimes, they are confused. Who wants to know this information? Why are they asking me this? We've had a couple of patients maybe say that they didn't want to meet with us, like, no I don't want to meet with you guys. Maybe they had a past experience with a social worker, and they didn't like it. Maybe they just didn't want to bring up some of those topics. I haven't seen that a lot. But yeah, we handle it pretty well, we introduce ourselves, we let them know why we want to, and patients are pretty open once they realize what's going on."* -Front line employee 9

Providers mentioned that housing was a particularly difficult resource to access given restrictions in DC around who is eligible for housing, and patients discussed long wait lists at those organizations for which they might be eligible.

### Domain 3: Inner setting

**Structural characteristics.** Barriers related to electronic health record (EHR) documentation of social risk factor screening and referral were discussed by many of the employees as an area for improvement. Employee participants suggested that they should be able to use a case management system within the EHR to identify patients and track patients with identified unmet social needs. Employees described a process of time-consuming tracking in excel spreadsheets, manual notes in the EHR, and direct communication between clinical staff. They described the shortcomings of the EHR system for outpatient inter-disciplinary case management.

*"... it would be nice if you could make folders within the notifications because doctors will send the message to me and that's how I get a social work consult pretty much on the outpatient side. And then I have to read through every message and before I read the message, the doctor can put a title to say what the subject matter is, but otherwise everybody's in one pool. So I'm not able to take some out and put it into a folder. This relates to all prenatal patients, women who are family planning, etc., and patients with social needs, patients with mental health needs...The inpatient side has a consult section because I work on the inpatient side, but the outpatient side is only a message box."* -Front line employee 11

The EHR further prevented linking children to the same parent and thus having social needs screenings and subsequent response linked between siblings and families.

*"...You can group families that have a particular way, but it's real labor intensive to put the families into the group and then everyone would have to do an actual referral for whatever the social determinants of health they did, even if it didn't really go anywhere in the EMR to then be able to link somehow what we were trying to jimmy the system, because otherwise it's all on an Excel spreadsheet. Who got referred for what? What was the disposition of the referral? So that also becomes so subject to, you know, trying to maintain updates on these referrals, especially when there are different people who have different roles in the various aspects of the referral. So a way to, and I'm not that group by any means, but I just wholeheartedly believe there's got to be a way for this to happen in the future."* -Front line employee 1

**Culture.** Participants largely deemed the organizational culture as supportive of efforts to address patient needs. Participants expressed that the institutional culture has improved over time, but some suggested that more could still be done to improve organizational culture.

*"So it's getting better... moving away from this kind of punitive, you're late, you can't be seen, you know, I'm going to roll my eyes at you kind of culture to a place where, you know, we just accept women where they are. And if they show up an hour late, you know, and they look a mess or whatever, we're just going to we're going to say we're going to respect them just like we would do anybody else. So, yeah, you know, I think there's just a piece of dignity in that... I think it's also all becoming more conscious of race and trauma and what's going on around us. I think it is helping we had a few trainings on trauma, informed care. . .I think all of that is helped."—Advanced practitioner employee 12*

*As an organization that not only is involved in medical and health related, who also is a large housing component, you know, I think staff and or staff members are kind of really aware of the struggles that, you know, a number of our patients are faced with . . . living in D.C. . . You know, I think that it's part of kind of our mission statement, an organizational structure to really kind of focus on. . .those aspects of a patient's life, not just the small health component with it. -Advanced practitioner employee 16*

Still, employees described ways in which the healthcare system had invested in providing resources to enable screening.

*"But one of the first things that kind of popped up in my head is the patient first model, which is striving to deliver the very best to every patient, every single day. And I think that really helps because, of course, for example, sometimes there can be a language barrier and I think [our healthcare system] has done a great job in offering interpreting services, not just language over the phone, but having physical deaf interpreters here. I personally have loved that concerning being able to screen every single patient and not let language be that barrier to not being able to do conduct those screens." -Front line employee 9*

**Implementation climate.** Overall, participants described receptivity to change, but also described practical constraints to delivering social risk factor screening and referrals. Participants further stated that their motivation to perform social risk factor screening and social need referrals stemmed from their desire to better serve the patient population.

*"My motivation is the success of the patient, if they met their goals, if their mental health has improved. some patients have dealt with some substance use and one patient was dealing with using PCP. My goal was trying to get her connected with these programs and things like that, but when I saw her weekly, she stopped using, that was a big thing for me. . ." -Front line employee 13*

*"I think we can I think one of the things with several things that we really have working for us . . . is we, we really do have a committed team who have clear buy in to the social determinants of health screening. The social determinants of health screening was never why do we have to do this? Is this our lane? That was never part of any discussion we ever had to have with any team member on our team. It's always how do we do it? How does it make more sense? You know, how do we communicate to families why we're doing it? And so I think that's why we've been able to have so many iterations of it, is because there's it's a dialogue, right? It's that's not working. Can we try this? . . . So there's this collective thought happening, like, are we getting at everything we need to get at to figure out how we can best help our families and understand what our families are dealing with." -Front line employee 1*

**Relative priority.**   Employees also described how both competing priorities and patient volume impacted prioritization of general questions for patients, as well as social risk factor screening or follow-up visits. For instance, grant funding prioritizing screening all new patients impacted the ability to follow up with those who had previously demonstrated high social need. Other considerations in prioritizing service delivery included addressing issues which are most likely to impact clinical outcomes, as well as those issues that could be intervened on.

*"... we have a case load of about over 250 patients because you know, they are supported throughout the pregnancy and up to a year after. So as far as prioritizing. Well. So as far as grants, the priority is catching those new OBs at their initial appointment, catching the postpartum appointments... So it's hard to juggle that and also catching patients who are just coming back for their regular return OB visits for their prenatal care that are being case managed, and we've had to prioritize seeing new OBs and not seeing our patients who are being case managed and hopefully following up with them via phone call." -Advanced practitioner employee 12*

**Readiness for Implementation.**   One of the elements that employee participants noted was limited available resources in clinic. In some of the included clinics patients were asked to contact CBOs directly and in others a social worker or navigator could assist in helping patients schedule appointments with CBOs. Some employees suggested co-location of a representative from commonly used referral services at clinic sites. Other employees desired more time to dedicate to following up with community organization on behalf of patients to better connect patients to needed resources and services.

*"I think in an ideal world, I would probably have our top five community resources, a representative from those top five within the clinic to automatically link patients with those people, because that would just be great to just have that knocked out, they could update patients about any changes. They can get them linked right there on the spot." -Front line employee 9*

*"What I would change is basically ... just giving the patient a referral, I would reach out myself, to be honest. That's the part that I would like to change, being able to connect the patient ourselves versus just giving the referral to the patient. I want us to be able to schedule, anything or get what we need to get for the patients so the patient is not doing that much work and we're able to just align them properly."–Front line employee 14*

Through the 5-year disparities initiative additional staff were being hired to integrate mental health services into clinics and leadership for the healthcare system were recommitting to providing additional resources to achieve the disparities project objectives, including staff as detailed in the descriptions of screening and referral costs.

## Domain 4: Characteristics of individuals

**Self-efficacy.**   Some providers, including medical residents and new patient coordinators, reported that they did not have training on social risk factor screening and how to address identified social need issues. Limited comfort sometimes led to skipping certain questions or struggling to figure out the appropriate pathways for patients with identified needs.

*I think it would be nice if we got training on it at the very beginning, just like on workflows...But just to have like a formalized set process that we tell all the interns, like, OK, when*

*you do the well child check Bright Futures form, this comes up positive. You know, you send a message to our social worker. And that's kind of established from the beginning. I think we just kind of picked that up as we go. We kind of learn about it. But just having some sort of training in terms of, you know, this is what we do every single time, you know, this question comes up positive, then, you know, you notify social work. . . What things need to be escalated to social work and how to do it, I think would be valuable.* -Advanced practitioner employee 5

Individuals who had worked in the clinics for at least a few years and those who were trained in social risk factor screening such as social workers largely felt confident in their ability to carry out social risk factor screenings.

*"So, you know, it's pretty easy to screen because, you know, you can go through those questions. And I feel pretty confident with being able to detect if there are some hesitant, they're hesitant to really give an answer. We can kind of, you know, take some time to ask more about that topic. So I've had it where, you know, they're a little hesitant to say anything about, like food insecurity, you know, and then you have to kind of ask more questions. But I feel pretty confident in that. . ."*- Advanced practitioner employee 5

Still, some employees recalled that skills in delivering screenings and identifying referrals were learned on the job, and that additional training would have made them more confident in administering such screenings and referrals.

When asked about their confidence in their ability to provide social risk factor referrals to patients most employee participants felt that they could provide some, but not all of the needed resources. In particular, housing instability was described as difficult for employees to intervene on given limited community resources and heightened housing needs from the pandemic.

*"Honestly, I don't have a lot of experience with pediatrics, but with adults, sometimes if they're having trouble with homelessness or housing insecurity, I feel like the resource that's given to them is just like a list of things to call. But often patients need more than just like this list. They need like walked through it because it's a very intimidating system. And I feel like that component is something that's not there."*–Advanced practitioner employee 8

*. . .but in D.C., you can pretty much only get housing resources if you have dependents. So if someone is pregnant for the first time or this is going to be the first baby that they will be parenting, whether that's by their own choice or by not, like they can't get housing resources in D. C. until they're until they are twenty-eight weeks.*–Front line Employee 15

### Domain 5: Process

**Executing.**  Most of the interviewees described challenges in implementing new processes. For instance, in the OB clinic, they delivered a multi-tool and multi-domain seven-page questionnaire to be completed during the visit, some patients were put off by the expectation that they would have a separate meeting with a navigator or social worker, as they had expected only a routine medical visit. This resulted in some participants leaving without knowing they were supposed to meet with behavioral health and others who did not want to stay after their medical visit. Some of the employee participants suggested a triage map listing of all steps to be completed during initial OB visits and could be sent to patients prior to arrival.

*"Yeah, I, I would say right now for my team, the biggest barrier would be the fact that patients do not know, that they'll be meeting with someone from the behavioral health team. What we've noticed at times that they usually, a normal new OB appointment is about an hour, just with the medical staff, whether it's the midwife, or any of the providers. And if we don't touch base with them prior to kind of let them know, like, hey, please don't leave we want to meet with you, they can you know, want to just get out of there after being there for an hour and then they still need to do blood work. I would say that's a very big barrier, patients not knowing that, I mean, it makes sense you're coming to your pre-natal appointment not to meet with the social worker."* -Frontline employee 9

While some providers mentioned translation services as a screening facilitator that supported a culture of inclusive screening, other employees discussed a need for additional translation services especially for referral resources, particularly in Spanish and Amharic.

Employee participants also discussed communication between employees (e.g. providers, administrative staff, nurses, social workers) as both a facilitator and barrier to providing social risk factor screening and referral services. Participants also described staffing shortages which lead to staff covering for roles or responsibilities that are outside the scope of their normal job responsibilities. Other barriers included limited avenues to coordinate notes and care between providers with siloed responsibilities.

*". . . the communication like in my experience between the behavioral health team and myself and I imagine other providers is not great. You know, we're getting different pieces of information and we may or may not know about some significant social determinants of health that the behavioral health team happens to pick up. Or, of course, if we're seeing them first, they're already out the door by the time we get a chance to talk to the behavioral health team. So communication is not great like in the moment just because we're all going in so many different directions. So one thing we've done to kind of help with that is to make it so that when the behavioral health person puts in the patient's responses, it will automatically populate into the clinical notes for the O.B. provider. And that is in process. . ."* -Advanced practitioner employee 12

The COVID-19 pandemic also significantly affected the referral process. Staff noted such rapid changes in community-based organization offerings that it was difficult to keep up with where to send patients who identified needs.

*"the biggest barrier. Well, there's covid. So policies are, oh my gosh, different agencies are changing their policies every day, especially concerning changes that may be happening in D. C. overall. So not being able to keep up with everything that every agency is offering and not offering has been a bit of an issue. Within that initial assessment, there may not be a lot of time to help call, every single place for that patient. . .But as time has gone by, we've been able to find out what agencies in the area are offering services, even new ones. . ."* -Front line employee 9

## Discussion

During the interview period social risk factor screening and social need referral processes were evolving in different ways across the specialty clinic sites due to both the COVID-19 pandemic and evolving workflow processes and staffing in relation to the 5-year initiative. Employee participants largely felt confident in their ability to carry out social risk factor screenings and

referrals but felt that additional resources in terms of time, training, care coordination between providers, and resources (including tracking systems) to monitor and address identified needs would be beneficial.

Previous studies highlighted similar organizational social risk factor screening facilitators to those discussed by our study participants such as increased time/staff available for screening and referral processes, use of social workers, and EHR modifications [29–33]. In previous studies, active referral processes through navigators or CBO representatives have been shown to increase use of referred resources compared to passive referrals of only providing information [32, 34–36]. Our findings corroborate that this active referral is preferable, but time and cost intensive. Previous studies have also reported limited tools at the organizational level. For instance, similar to our findings, a 5-site study among federally qualified health centers in Michigan reported inconsistent screening integration in the EHR, significant variation in screening processes by site, and barriers including resource and staff availability, and competing priorities [37]. A implementation study in 12 primary care practices in Virginia also highlighted the intensive nature of social needs screening in standard care and challenges in providing resources to address identified needs [38]. These challenges of balancing time and resources needed for screening and referral will remain unless, among other factors, payment models change to provide coverage for the provided services.

The need to better track social needs screening and referral has also been widely documented and is currently a priority in local and national projects. For instance, across the country through the federal government's Office of the National Coordinator, initiatives such as the Gravity Project, and with collaborations between the American Medical Association and United Healthcare, resources are being invested both in standardization of interpreting screeners and assigning ICD-10 codes [39–41]. Within DC through the Community Resource Information Exchange (CoRIE) and even within some of the clinics interviewed, there are health information exchange projects focused on standardizing coding of social needs and improving closed loop referral systems for both clinical and research purposes. A recent systematic review highlighted the growing body of studies on integrating social needs screening into the EHR and cited implementation challenges, as well as limited effects on health and health care cost and utilization [42]. At the time of our study, referral tracking was limited by lack of EHR integration and not having all clinics assign ICD-10 codes to identified needs. In one published study, EHR automated solutions for referral tracking have been used effectively in other situations such as converting fax referrals to EHR automated referrals and integrating automatic ICD-10 codes based on social risk factor questionnaire responses [43, 44], but additional work is needed to better understand best strategies for integration both in our project and more generally.

At the individual level there was acknowledgement of the need to include patients in planning of social needs screening and referral processes. Another study in a healthcare system showed that patients largely support asking about and addressing social needs in a medical setting, particularly if they had experienced such need [45]. Still, only a fraction of those with needs identified wanted help in addressing their social risk factors. Furthermore, a previous review of social risk factor screening barriers and facilitators discussed the importance of patient-centered care, similar to the employee participants in our study who cited the importance of adaptability to meet patients where they are [46].

Limited studies have looked at the impact of social risk factor screening and referral on resolution of health-related social needs. A 2016 review of 39 published studies on the impact of investments in social services (housing, income, nutrition support, care coordination) on healthcare outcomes reported that 32 of the studies reported a positive effect on health outcomes, costs, or both [47]. One published study on referral completion reported that increasing the number of outreach encounters and follow-up time were associated with successful

referrals in an urban pediatric practice [48]. Programs supporting social risk factor screening implementation have been shown to increase use of referred resources in family medicine, pediatric, women's health, and oncology settings [29–33]. Additional work describes the potential to affect access to quality of care via social risk factor screening [49]. The large, Accountable Healthcare Communities project from the Center for Medicare and Medicaid Services, focused on reducing 30-day hospital readmission reported that 74% of eligible beneficiaries accepted navigation, but only 14% of those who completed a year of navigation had documentation that social needs were resolved [50]. This underscores the challenges of adequately resolving social needs even with dedicated resources for screening and follow-up. Future studies should further assess adaptation of screening tools in relation to effectiveness of screening and referral on resolution of social needs or relevant health outcomes. Understanding these outcomes will be critical to determining payment models and creating sustainable screening and referral systems.

Strengths of our study including using a highly-utilized theoretical framework to understand organizational level impacts on social risk factor screening and referral. The CFIR has been applied to screening and referral processes in alcohol abuse and treatment, cancer, chronic disease, primary care, and school-based sexual health settings [51–55]. Contextualizing screening and referral processes through a widely utilized framework allows for comparison with other clinical setting findings. Other strengths of our study include including diverse stakeholders in interviews to better describe contextual factors affecting social risk factor screening and referral across multiple clinics within a health-disparities focused project. While conducting the project during a pandemic led to a dynamic intervention process, it also enabled us to capture changes as they were happening and to describe adaptions to the screening and referral interventions.

## Limitations

One of the major limitations of this study was the inability to conduct interviews in person due to the COVID-19 pandemic. We also were unable to conduct any clinical workflow observations due to the pandemic. This was especially challenging for patient interviews which were difficult to schedule and were largely conducted by phone instead of video. However, patients interviewed from home may have felt more comfortable discussing their experiences with social risk factor screening and referral processes. Furthermore, our sample was recruited using convenience sampling where both the employee and patient participants were suggested by clinical leads rather than sampled at random, which could have led to bias in selection. Additionally, we only interviewed English speaking patients and non-English speaking patients may face different barriers and facilitators which were not examined in our study; future work is needed to understand barriers for non-English speaking populations. Limited quantitative data on social needs and tracking referrals due to evolving tracking methods also prevented us from triangulating our qualitative findings with quantitative data. We plan to triangulate our findings with quantitative data once social needs data is better captured within the EHR and once screening adaptations are in place to assess how changes affect screening and referral results. Current modifications as part of the 5-year initiative include standardizing social risk factor responses and integrating externally collected forms into the medical record, thus making it easier to synthesize the quantitative data.

## Conclusion

This qualitative study describes the processes, strengths, and challenges of social risk factor screening and referral processes in early implementation across multiple specialty clinics

related to maternal and infant health. Implementation of social risk factor screening and impact on health outcomes is still an emerging area in the literature. Future studies, including this project, may be able to measure how identification and intervention on unmet social needs affects intermediate outcomes and subsequently can reduce disparities in maternal and infant mortality within D.C., particularly in high-risk populations. Programmatically, these findings may be used to address identified screening and referral barriers and to facilitate better tracking to better understand burden of need and ability to resolve identified issues. Next steps include testing potential solutions to identified barriers to screening and referral implementation. Future studies should go further to demonstrate how integration of screening and referral sources affect specific health outcomes, healthcare utilization, and patient satisfaction, in order to make the case to payers that reimbursement for these services is needed.

## Supporting information

**S1 Table. Domains covered in administered questionnaires.**
(DOCX)

## Acknowledgments

The authors would like to thank the following individuals who contributed: Kelly S. McShane, Justin M. Hughes, Christine D. Laccay, Michelle A. Roett. We would also like to thank the interview participants for sharing their experiences, opinions, and feedback with the research team.

## Author Contributions

**Conceptualization:** Angela Thomas, Hannah Arem.

**Data curation:** Hannah Arem.

**Formal analysis:** Jason Brown, Naheed Ahmed, Hannah Arem.

**Funding acquisition:** Angela Thomas.

**Investigation:** Naheed Ahmed, Hannah Arem.

**Methodology:** Hannah Arem.

**Project administration:** Jason Brown, Naheed Ahmed, Hannah Arem.

**Software:** Hannah Arem.

**Supervision:** Hannah Arem.

**Writing – original draft:** Jason Brown, Hannah Arem.

**Writing – review & editing:** Naheed Ahmed, Matthew Biel, Loral Patchen, Janine Rethy, Angela Thomas, Hannah Arem.

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
