## [Decision Letter · Decision Letter 0]

18 Aug 2022

PONE-D-22-00415Considerations in implementation of social risk factor screening and referral in maternal and infant care in Washington, DC: a qualitative studyPLOS ONE

Dear Dr. Arem,

Thank you for submitting your manuscript to PLOS ONE. After careful consideration, we feel that it has merit but does not fully meet PLOS ONE’s publication criteria as it currently stands. Therefore, we invite you to submit a revised version of the manuscript that addresses the points raised during the review process. In general, I agree with the reviewers comments. I would pay particular attention to describing who participate/included and who was not. Include response rates for all levels of staff and patients. It would also be helpful if the authors provided a descriptor of the employee with the representative quote (e.g., Employee -11, Senior Leadership). Finally, add in a discussion about who was represented in the study and the implications of representativeness in interpreting the results, generalizability, and future research. 

We look forward to receiving your revised manuscript.

Kind regards,

Kristina Hood, Ph.D.

Academic Editor

PLOS ONE

Journal Requirements:

Reviewers' comments:

Reviewer's Responses to Questions

**Comments to the Author**

1. Is the manuscript technically sound, and do the data support the conclusions?

Reviewer #1: No

Reviewer #2: Yes

Reviewer #3: Yes

2. Has the statistical analysis been performed appropriately and rigorously? 

Reviewer #1: No

Reviewer #2: Yes

Reviewer #3: Yes

3. Have the authors made all data underlying the findings in their manuscript fully available?

Reviewer #1: Yes

Reviewer #2: No

Reviewer #3: Yes

4. Is the manuscript presented in an intelligible fashion and written in standard English?

Reviewer #1: No

Reviewer #2: No

Reviewer #3: Yes

5. Review Comments to the Author

Reviewer #1: I appreciate all your work but believe you need to address some significant issues in order to receive further publication consideration. My comments follow, not in any order of priority:

1.Abstract: results section seems too long.

2.Bacjground: I suggest you start with a more expanded paragraph on the problem with worldwide. U. S., and D C data on prevalence and consequences. That seems the best starting point. Then you can go to social determinants as "causes" or key factors.

3.Next you need a section titled Literature Review (LR) and a professional LR to establish your study addresses a gap in the LR. The LR should include: Research questions/hypotheses; databases and keywords used; time period covered by review; inclusion/exclusion criteria for selecting articles; total articles found; total articles reviewed; major findings , including named gaps.

4.Before Methods you need IRB approval, funding source, any researcher conflicts of interest, where and when the study was done.

5.Methods: You need citations to the five year project and when it began and current status. CBOs: spell out the full term the first time you use it.

6. Replace STUDY POPULATION with SAMPLE and SAMPLING PROCESS and better state the sample includes two subsets--employees and patiets and then for each, a better description of how you selected them--random? Seems it is convenience sampling which you should state and note also in a limitations section which you need at the end of the paper prior to any discussion.

7.Data Analysis: How did the coders resolve any differences? How did they move from coding to themes? Was anyone else involved?

8.Results: You need a subheading each for Screening and one for Referral, before the narratives on each. I actually am jot clear how relevant these results our to your core research questions. You might consider deleting them and having a very brief statement on screening and referral in the new Sampling section suggested above.

9. Results: Quantify instead of using words like Many or high proportion.

10. Results.: I suggest you have one results section on Patients; one on employees and one combined. The current format does not systematically allow an overall view and a view of each sample's perspectives.

11.As noted above, you need a Limitations section before your Discussion section.

Reviewer #2: This was a compelling and well-written manuscript that used an innovated methodology to assess various perspectives on social needs screenings. The authors summarized their findings in a clear and concise way, allowing the reader to easily identify key points throughout. The manuscript is also strengthen by the populations represented--rather than only recruiting health care team members, including both the patient and CBO perspectives offers a more holistic understanding of the current state and opportunities for improvement.

Because the sample size is small (which is not a limitation for this qualitative study), it would be helpful to understand more about who is included and who is not. Specifically, understanding response rates and whether the research team was able to achieve its goals of recruiting all levels of staff from each site, and whether more than 3 patients were recruited (and how many declined), would be helpful information at the beginning of the findings. Table 2 provides much of this information but it would be helpful to have some information about response rates in the body of the manuscript before findings are presented. In terms of who is not represented, it would also be worth the researchers pointing to the limitations of only recruiting English-speaking patients and potentially including a recommended next step of understanding the experiences of non-English-speaking patients in future research.

This is a strong study that is ripe for publication given its methodological soundness and representation from multiple perspectives.

Reviewer #3: This study presents important qualitative findings from using a multi-stakeholder design to identify the facilitators and barriers associated with social risk screening and referral processes within a single health system in a large urban setting.

Intro

In the sentence of paragraph 1 that refers to social needs, it may help to clarify socials needs as “…those social risk factors for which patients want help to address…”

Methods

In the Study Population section, in the 2nd paragraph, 1st sentence that describes the project leads, it refers to three levels, though there are only two listed: administrative and clinical.

A diagram of the interview participant matrix would be helpful, to understand the various levels, clinics, specialties, and partner organizations that compose the sampling frame.

In the Study Population section, in the 3rd paragraph: please clarify what type of screening for which the patient had to be eligible.

For recruitment, please describe how many invites were sent and how many interviews completed. This will provide a response rate.

It may be helpful to detail the social risk/need domains that were considered as part of this study.

Results

Adding some sub-headings to the first three paragraphs, as is done for the remainder of the results may help with clarity for readers.

The results cover a wide range of domains/themes that are important.

Would it be possible to list the role or department of the participant, instead of the participant ID (i.e., Employee 1)? It seems that this information may offer some context to support the response/quote.

Another limitation to note is the focus on English-speaking participants

Discussion

The discussion nicely places the findings within the current literature and offers suggestions for future studies in this field

6. PLOS authors have the option to publish the peer review history of their article (what does this mean?). If published, this will include your full peer review and any attached files.

Reviewer #1: **Yes: **Dr. William Cabin

Reviewer #2: **Yes: **Courtnee Hamity

Reviewer #3: No

---

## [Author Response · Author response to Decision Letter 0]

30 Sep 2022

Dear Dr. Hood,

We would like to thank you and the reviewers for the careful consideration of our manuscript PONE-D-22-00415 entitled “Considerations in implementation of social risk factor screening and referral in maternal and infant care in Washington, DC: a qualitative study.” The thoughtful comments and constructive criticism have helped to improve our manuscript. Our point-by-point responses to reviewer comments are in bold below with line numbers of edits included where necessary. Revisions were tracked and line numbers reflect lines from the revised text. 

Thank you for the consideration of our revised manuscript. 

Editor 

I would pay particular attention to describing who participate/included and who was not. Include response rates for all levels of staff and patients. We have added this description on pages 4-5 separately for providers and patients.

 It would also be helpful if the authors provided a descriptor of the employee with the representative quote (e.g., Employee -11, Senior Leadership). We did not include specific role as it would provide sufficient information within the institution to identify the participant. However, we did add the “level” of the participants to provide more specificity, while protecting participant identity.

Finally, add in a discussion about who was represented in the study and the implications of representativeness in interpreting the results, generalizability, and future research. We have added a sentence in the limitations “Furthermore, our sample was recruited using convenience sampling where both the employee and patient participants were suggested by clinical leads rather than sampled at random, which could have led to bias in selection.”

Reviewer #1: 

1.Abstract: results section seems too long. We have made minor edits to condense the abstract results section but wanted to include key findings within the limits of journal word count specifications. 

2.Background: I suggest you start with a more expanded paragraph on the problem with worldwide. U. S., and DC data on prevalence and consequences. That seems the best starting point. Then you can go to social determinants as "causes" or key factors. We appreciate the need to add more background on prevalence and consequences. We have added some specific statistics for the US and DC on lines 67-70. 

3.Next you need a section titled Literature Review (LR) and a professional LR to establish your study addresses a gap in the LR. The LR should include: Research questions/hypotheses; databases and keywords used; time period covered by review; inclusion/exclusion criteria for selecting articles; total articles found; total articles reviewed; major findings, including named gaps. We appreciate the importance of a systematic literature review in this area, but believe that this process merits its own paper and is beyond the scope of the background section in this paper. Still, we have added a citation to a recent systematic review covering screening and referral for unmet social needs to better describe the existing landscape in this area and have added additional references to the discussion. 

4.Before Methods you need IRB approval, funding source, any researcher conflicts of interest, where and when the study was done. We have moved the details regarding IRB approval and added details regarding the timeframe of study (see lines 104-107). The competing interest statement has been moved as requested. Per journal instructions, funding source is not included in the body of the manuscript. 

5.Methods: You need citations to the five-year project and when it began and current status. CBOs: spell out the full term the first time you use it. While the study protocol has not yet been published, we added a web page citation to the project website at https://www.medstarhealth.org/services/dc-safe-babies-safe-moms. We also reviewed acronym usage throughout the document and CBOs were identified in it’s first use on line 101 of the revised document. 

6. Replace STUDY POPULATION with SAMPLE and SAMPLING PROCESS and better state the sample includes two subsets--employees and patients and then for each, a better description of how you selected them--random? Seems it is convenience sampling which you should state and note also in a limitations section which you need at the end of the paper prior to any discussion. We have added clarity to sample and sampling process in methods and limitations sections as suggested. 

7.Data Analysis: How did the coders resolve any differences? How did they move from coding to themes? Was anyone else involved? We appreciate the reviewers comment and need for clarity here. Details were added regarding coding differences and conversion of codes to themes (see lines 171-175). 

 8.Results: You need a subheading each for Screening and one for Referral, before the narratives on each. I actually am not clear how relevant these results to your core research questions. You might consider deleting them and having a very brief statement on screening and referral in the new Sampling section suggested above. We have added additional headers for the screening and referral sections along with further edits to the results section. 

9. Results: Quantify instead of using words like Many or high proportion. While we appreciate the importance of precision in summarizing results, given the qualitative nature of this study we believe that a qualitative description is more appropriate than providing a quantitative summary of how many people responded to identify certain CFIR themes. In a future larger sample, we agree that it would be interesting to conduct a survey and quantify these results.

10. Results.: I suggest you have one results section on Patients; one on employees and one combined. The current format does not systematically allow an overall view and a view of each sample's perspectives. We appreciate the suggestion from reviewer 1 and have tried to increase clarity of the results sections with additional headers and edits. We moved the patient comments to the process section of the results.

11.As noted above, you need a Limitations section before your Discussion section. A subheading was added to better clarify this section prior to the conclusion. 

Reviewer 2

Because the sample size is small (which is not a limitation for this qualitative study), it would be helpful to understand more about who is included and who is not. Specifically, understanding response rates and whether the research team was able to achieve its goals of recruiting all levels of staff from each site, and whether more than 3 patients were recruited (and how many declined), would be helpful information at the beginning of the findings. We appreciate the need to add this information and response rates for both patient and employee interviews have been added (see lines 128-131 and 139-144). 

Table 2 provides much of this information but it would be helpful to have some information about response rates in the body of the manuscript before findings are presented. In terms of who is not represented, it would also be worth the researchers pointing to the limitations of only recruiting English-speaking patients and potentially including a recommended next step of understanding the experiences of non-English-speaking patients in future research. We added additional limitations regarding interviewing English-speaking patients. 

This is a strong study that is ripe for publication given its methodological soundness and representation from multiple perspectives. We appreciate Reviewer 2’s feedback regarding methodologic design and our efforts to have a multi-stakeholder sample.

Reviewer 3

Intro

In the sentence of paragraph 1 that refers to social needs, it may help to clarify socials needs as “…those social risk factors for which patients want help to address…” We have added the clarification of unmet social needs to the sentence in question. 

Methods

In the Study Population section, in the 2nd paragraph, 1st sentence that describes the project leads, it refers to three levels, though there are only two listed: administrative and clinical. We have corrected the text to reflect the two levels. 

A diagram of the interview participant matrix would be helpful, to understand the various levels, clinics, specialties, and partner organizations that compose the sampling frame. We appreciate the suggestion to change Table 2 to a figure/diagram and have converted Table 2 to a figure that hopefully is more helpful in understanding our study sample. 

 In the Study Population section, in the 3rd paragraph: please clarify what type of screening for which the patient had to be eligible. We have edited the text to clarify that patients were eligible for SDoH screening (see lines 132-133). 

For recruitment, please describe how many invites were sent and how many interviews completed. This will provide a response rate. We appreciate the reviewers highlighting the need to add this information and response rates for both patient and employee interviews have been added (see lines 129-131 and 139-144). 

It may be helpful to detail the social risk/need domains that were considered as part of this study. We have added a supplemental table 1 to show the domains covered in the actual questionnaires, although other issues were frequently uncovered in discussion.

 Results

Adding some sub-headings to the first three paragraphs, as is done for the remainder of the results may help with clarity for readers. We have added sub headings to first three paragraphs as suggested along with further edits to results section. 

Would it be possible to list the role or department of the participant, instead of the participant ID (i.e., Employee 1)? It seems that this information may offer some context to support the response/quote. We appreciate the reviewer comment about more descriptive participant labels. However, we did not want to include too much information to protect the privacy/anonymity of participants, specifically for the employees. Thus, we added clinical or administrative levels but chose not to be more descriptive than that. 

Another limitation to note is the focus on English-speaking participants. We added a sentence to the limitations regarding interviewing only English-speaking patients.

---

## [Decision Letter · Decision Letter 1]

20 Mar 2023

Considerations in implementation of social risk factor screening and referral in maternal and infant care in Washington, DC: a qualitative study

PONE-D-22-00415R1

Dear Dr. Arem,

We’re pleased to inform you that your manuscript has been judged scientifically suitable for publication and will be formally accepted for publication once it meets all outstanding technical requirements.

Kind regards,

Kristina Hood, Ph.D.

Academic Editor

PLOS ONE

Additional Editor Comments (optional):

Reviewers' comments:

Reviewer's Responses to Questions

**Comments to the Author**

1. If the authors have adequately addressed your comments raised in a previous round of review and you feel that this manuscript is now acceptable for publication, you may indicate that here to bypass the “Comments to the Author” section, enter your conflict of interest statement in the “Confidential to Editor” section, and submit your "Accept" recommendation.

Reviewer #3: All comments have been addressed

Reviewer #4: All comments have been addressed

2. Is the manuscript technically sound, and do the data support the conclusions?

Reviewer #3: Yes

Reviewer #4: Yes

3. Has the statistical analysis been performed appropriately and rigorously? 

Reviewer #3: N/A

Reviewer #4: Yes

4. Have the authors made all data underlying the findings in their manuscript fully available?

Reviewer #3: Yes

Reviewer #4: No

5. Is the manuscript presented in an intelligible fashion and written in standard English?

Reviewer #3: Yes

Reviewer #4: Yes

6. Review Comments to the Author

Reviewer #3: The authors have done a great job revising the manuscript based on reviewer feedback. The manuscript is quite rich with information, reflecting the value of qualitative research, and offers great insight into the processes associated with social risk screen/referral programs within a health care setting. The participant quotes offer excellent context. The authors provide a direction for future research, which is exactly what is needed on this scientific topic.

Reviewer #4: This study, guided by the CFIR implementation framework, identified processes and organizational factors that were barriers or facilitators of implementation of screening and referral for social needs within maternal and infant care clinics in Washington, DC. This is a well-written manuscript that presents in-depth qualitative data on a topic of high national interest: social needs screening and referral to increase health equity. The findings of the paper are important, methods and analysis are rigorous, and conclusions follow logically from results.

The authors appear to have been thoughtful and thorough in their response to the editor’s and reviewers’ prior comments. I find this revision to be polished and ready for publication.

My only recommendation is that the authors consider whether they have or can follow the reporting standards from one of the two primary checklists for reporting qualitative research: COREQ (https://www.equator-network.org/reporting-guidelines/coreq/) or SRQR (https://www.equator-network.org/reporting-guidelines/srqr/). I think the authors already report all or nearly all of the checklist items. If possible, it would strengthen the paper to fulfill criteria for or the other and cite the checklist in the paper.

Minor comments:

1. Just a thought for the authors to consider: I don’t think the inability to conduct interviews in person (due to COVID) was necessarily a limitation. As exemplified by this manuscript and many recent mixed-methods publications, virtually conducted interviews yield rich data and likely increase access and inclusion for diverse study participants.

2. Perhaps I missed something, but in the abstract the authors state they conducted interviews at 3 clinics and one community organization, and later 4 clinics are mentioned. Worth a double check that numbers are consistent.

7. PLOS authors have the option to publish the peer review history of their article (what does this mean?). If published, this will include your full peer review and any attached files.

Reviewer #3: No

Reviewer #4: No

---

## [Editor Report · Acceptance letter]

3 Apr 2023

PONE-D-22-00415R1 

Considerations in implementation of social risk factor screening and referral in maternal and infant care in Washington, DC: a qualitative study 

Dear Dr. Arem:

I'm pleased to inform you that your manuscript has been deemed suitable for publication in PLOS ONE. Congratulations! Your manuscript is now with our production department. 

Kind regards, 

on behalf of

Dr. Kristina Hood 

Academic Editor

PLOS ONE